# Sequential Deconfounding for Causal Inference with Unobserved Confounders

## Abstract

Observational data is often used to estimate the effect of a treatment when randomized experiments are infeasible or costly. However, observational data often yields biased estimates of treatment effects, since treatment assignment can be confounded by unobserved variables. A remedy is offered by deconfounding methods that adjust for such unobserved confounders. In this paper, we develop the Sequential Deconfounder, a method that enables estimating individualized treatment effects over time in presence of unobserved confounders. This is the first deconfounding method that can be used with a *single* treatment assigned at each timestep. The Sequential Deconfounder uses a novel Gaussian process latent variable model to infer substitutes for the unobserved confounders, which are then used in conjunction with an outcome model to estimate treatment effects over time. We prove that using our method yields unbiased estimates of individualized treatment responses over time. Using simulated and real medical data, we demonstrate the efficacy of our method in deconfounding the estimation of treatment responses over time.

## 1 Introduction

Individual-level decision-making is of increasing importance in domains such as marketing (Brodersen et al., 2015), education (Mandel et al., 2014), and medicine (Hill and Su, 2013). For instance, in the medical domain, treatment decisions are based on the individual response of a patient to a treatment and must be adapted over time according to the diseases progress. Hence, physicians must know the treatment effect in order to decide how to treat a patient. Since randomized experiments are often costly or otherwise infeasible, several methods have been proposed to estimate treatment effect over time from observational data (e.g., Lim et al., 2018; Robins and Hernán, 2009). These methods are largely based on the assumption that there are no unobserved confounders, i.e., variables that affect both the treatment assignment and the outcome. In practice, however, this assumption generally fails to hold, which yields biased estimates and, therefore, may invalidate any conclusions drawn from such methods.

We discuss how unobserved confounders introduce bias based on the following example. Consider a physician in a hospital that prescribes medications (i.e., treatments) to a patient, whereby the outcome is a measurement of a disease (e.g., a lab test). In this setting, there are many potential confounders, which affect both the treatment assignment and the patient outcome. For instance, treatment assignment is often confounded by social and behavioral data such as socio-economic status or housing conditions. Although recognized as important, social and behavioral data are routinely not captured in electronic health records (EHRs) (Adler and Stead, 2015; Cantor and Thorpe, 2018; Chan et al., 2010). Since these confounders are unobserved, we cannot control for them when estimating the effect of a treatment. And, if not controlled for, these confounders introduce statistical dependence between treatment assignment and patient outcome, which yields biased estimates. Hence, methods are required that can estimate treatment effects in presence of such unobserved confounders.

In order to cope with unobserved confounders, a theory for deconfounding (i.e., adjusting for unobserved confounders) was developed in the static setting, which assumes that there are multiple treatments available (Wang and Blei, 2019a). Bica et al. (2020b) extended this theory to a specific sequential setting. They infer

latent variables that act as substitutes for the unobserved confounders using a factor model. However, in order to infer latent variables, their theory requires multiple (i. e., *two* or more) treatments at each timestep. However, the theory for multiple treatments is not applicable to the common setting with a *single* treatment. The reasons for this are two-fold: First, existing works assume "(sequential) single strong ignorability" (Wang and Blei, 2019a; Bica et al., 2020b), which requires that, at every timestep, the unobserved confounder influences at least two treatments. In case there is only a single treatment, this assumption holds never true. Second, existing works leverage the dependency between multiple treatments to infer a substitute confounder (via a factor model). However, in the case of a single treatment, there is no dependency between different treatments that could be leveraged. Because of these reasons, the theory of deconfounding in the multiple treatments settings *cannot* be used in the common setting with a *single* treatment at each timestep.

In contrast to this, we develop a theory for deconfounding the estimation of treatment responses over time that does not rely on multiple treatments at each timestep; instead, our theory can be used in the common case in which there is a single treatment available at each timestep. In particular, instead of the "(sequential) single strong ignorability" which is assumed by existing works and never true in our setting, our work assumes "time-invariant unobserved confounding", which requires the unobserved confounder to be constant in time. Moreover, instead of leveraging the dependence between multiple treatments at each timestep, which is prohibited in our setting, we leverage the sequential dependence between the treatments over time to infer substitutes for the unobserved confounders. This requires a different model than a factor model. To the best of our knowledge, this is the first work that exploits the sequential dependence between treatment assignment to enable unbiased estimation of treatment responses over time in presence of unobserved confounders.

In this paper, we propose the Sequential Deconfounder, a method for estimating treatment responses over time in presence of unobserved confounders. Our method proceeds in two steps: (i) We fit a specific latent variable model that captures the sequential dependence between assigned treatments. By doing so, we obtain latent variables, which act as substitutes for the unobserved confounders. (ii) We control for the substitutes and obtain estimates of the treatment responses via an outcome model. We prove that this yields unbiased estimates of individualized treatment responses over time. Informed by this theoretical result, we propose an instantiation of the Sequential Deconfounder based on a novel Gaussian process latent variable model. Finally, we demonstrate the efficacy of the Sequential Deconfounder using both simulated data, where we can control the amount of confounding, and real-world medical data.

We summarize our **contributions**[1] as follows:

1. We develop a theory for deconfounding the estimation of treatment responses in presence of unobserved confounders when a single treatment is assigned sequentially, i. e., over time.

2. We propose, based on this theory, the Sequential Deconfounder, a method that infers substitutes for the unobserved confounders and yields unbiased estimates of individualized treatment responses over time.

3. We provide an instantiation of the Sequential Deconfounder via a novel Gaussian process latent variable model. The performance of state-of-the-art algorithms is substantially improved by our method.

## 2 Related Work

Extensive work focuses on estimating treatment effects in the static setting (e. g., Alaa and van der Schaar, 2017; Johansson et al., 2016; Shalit et al., 2017; Wager and Athey, 2018). In contrast, our work focuses on estimating individualized treatment responses over time, i. e., we consider sequences of treatments. In the following, we discuss (i) works on estimating treatment responses over time assuming no unobserved confounders and (ii) works on deconfounding the estimation of treatment effects.

**(i) Treatment responses over time.** Methods for estimating responses to a sequence of treatments originate primarily from the epidemiology literature. Among these methods are *g*-computation, structural

---

[1]Code available at https://github.com/anonymous/SeqDeconf (Link anonymized for peer-review; code for review in the supplements.)

nested models, and, in particular, marginal structural models (e. g., Robins, 1986; Robins and Hernán, 2009; Robins et al., 2000). These methods have been extended using recurrent neural networks and adversarial balancing (Lim et al., 2018; Bica et al., 2020a). In order to incorporate uncertainty quantification, Gaussian processes have been tailored to the estimation of treatment responses in the continuous-time setting (Schulam and Saria, 2017; Soleimani et al., 2017). The aforementioned methods have found widespread use; however, without exception, all of these methods rely on the assumption that there are no unobserved confounders. In practice, this assumption rarely holds true such that estimates obtained from these methods can be biased.

Our work addresses this shortcoming by developing a method for deconfounding, which can be used in conjunction with the above approaches and can lead to unbiased estimates of treatment responses over time.

**(ii) Deconfounding the estimation of treatment effects.** Deconfounding uses latent variable models to adjust for unobserved confounders. One possible approach is to rely proxies of the unobserved confounder to infer latent variables that are used for adjustment (Louizos et al., 2017; Lu et al., 2018; Witty et al., 2020). However, proxies may often not be available in practice. Instead, there have been several attempts to infer unobserved confounders in the static setting only from the treatment assignment (e. g., Tran and Blei, 2017; Wang and Blei, 2019a; Wang et al., 2018; Zhang et al., 2019). These methods estimate treatment effects in the static setting and assume that there are multiple treatments assigned. In particular, Wang and Blei (2019a) leverages the dependence between multiple treatments to infer substitutes for the unobserved confounders using a factor model. The only approach in the sequential setting is an extension of Wang and Blei (2019a) to a specific sequential setting, that is, when there are multiple treatments available at each timestep (Bica et al., 2020b). Similar to Wang and Blei (2019a), multiple treatments at each timestep are assumed to infer latent variables via a factor model. This factor model leverages the dependence between the multiple treatments at each timestep in order to infer substitutes for the unobserved confounders at each timestep. Both theory and methods *entirely* rely on the assumption of multiple treatments and, hence, none of the above methods can be applied in the common sequential setting with a *single* treatment assigned at each timestep.

As a remedy, we develop a theory for sequential deconfounding which can be applied in the common sequential setting with a single treatment at each timestep.

## 3  Problem Setup

In this section, we introduce the setup and notation used to study treatment responses and formalize the problem of unobserved confounders. For this, we consider a patient $i$, for which the random variables $X_t^{(i)} \in \mathcal{X}_t$ are the observed covariates (e. g., blood pressure), a single $A_t^{(i)} \in \mathcal{A}_t$ assigned at time $t$, and $Y_{t+1}^{(i)} \in \mathcal{Y}_t$ are the observed outcomes. Static covariates (e. g., genetic information) are part of the observed covariates. Note that we do not assume multiple treatments at each timestep. We consider the common case in which there is a single treatment available at each timestep.

Observational data on patient trajectories consists of $N$ independent realizations of the above variables for $t = 1$ until $T^{(i)}$, i. e., $\mathcal{D} = (\{x_t^{(i)}, a_t^{(i)}, y_{t+1}^{(i)}\}_{t=1}^{T^{(i)}})_{i=1}^N$. For simplicity, we omit the patient superscript $(i)$.

We introduce further notation as follows. Let $\bar{X}_t = (X_1, \ldots, X_t) \in \bar{\mathcal{X}}_t$ denote the history of covariates and let $\bar{A}_t = (A_1, \ldots, A_t) \in \bar{\mathcal{A}}_t$ denote the treatment history up to timestep $t$. Realizations of these random variables are denoted by $\bar{x}_t$ and $\bar{a}_t$, respectively.

We build upon the potential outcomes framework (Rubin, 1978; 2005), which was extended to take into account sequences of treatments (Robins and Hernán, 2009). Let $Y(\bar{a}_t)$ be the potential outcome for the treatment history $\bar{a}_t$. If $\bar{a}_t$ coincides with the treatment history in the data, then the outcome is observed. Otherwise, the outcome is not observed. For each patient, we aim to estimate the individualized treatment response, i. e., the outcome conditional on patient covariate history: $\mathbb{E}[Y(\bar{a}_t) \mid \bar{X}_t = \bar{x}_t]$.

For this, we make two standard assumptions (Robins and Hernán, 2009; Schulam and Saria, 2017):

**Assumption 1.** *(Consistency.) If $\bar{A}_t = \bar{a}_t$ for a given patient, then $Y(\bar{a}_t) = Y$ for that patient.*

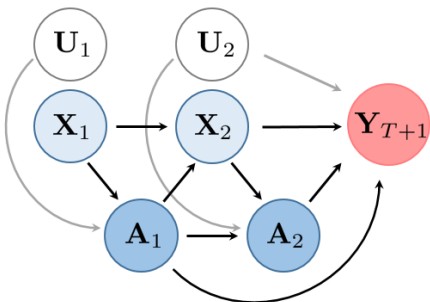

Figure 1: An illustration of the setup. A patient with covariates $X_t$ is assigned a treatment $A_t$. In practice, the assigned treatment and the patient's outcome are often confounded by an unobserved confounder $U_t$, which prohibits the use of standard methods for estimating treatment effects over time. The variables can be connected to the outcome in an arbitrary way.

**Assumption 2.** *(Positivity.) If* $\mathbb{P}(\bar{A}_{t-1} = \bar{a}_{t-1}, \bar{X}_t = \bar{x}_t) \neq 0$ *then* $\mathbb{P}(\bar{A}_t = \bar{a}_t \mid \bar{A}_{t-1} = \bar{a}_{t-1}, \bar{X}_t = \bar{x}_t) > 0$ *for all* $\bar{a}_t \in \bar{\mathcal{A}}_t$.

When using observational data, we can only obtain estimates of $\mathbb{E}[Y \mid \bar{A}_t = \bar{a}_t, \bar{X}_t = \bar{x}_t]$, but not for $\mathbb{E}[Y(\bar{a}_t) \mid \bar{X}_t = \bar{x}_t]$. However, many existing methods assume that these quantities are equal by assuming *sequential strong ignorability*:

$$Y(\bar{a}_t) \perp\!\!\!\perp A_t \mid \bar{A}_{t-1}, \bar{X}_t, \tag{1}$$

for all $\bar{a}_t \in \bar{\mathcal{A}}_t$ and for all $t = 1, \dots, T$. Sequential strong ignorability holds if there are no unobserved confounders $U_t$. If this condition held true, then $\mathbb{E}[Y(\bar{a}_t) \mid \bar{X}_t = \bar{x}_t]$ could be estimated from observational data, since $\mathbb{E}[Y(\bar{a}_t) \mid \bar{X}_t = \bar{x}_t] = \mathbb{E}[Y \mid \bar{A}_t = \bar{a}_t, \bar{X}_t = \bar{x}_t]$. In practice, however, sequential strong ignorability may fail to hold, since there exist unobserved confounders $U_t$.

Hence, individualized treatment responses cannot be estimated from observational data using standard methods, since they estimate $\mathbb{E}[Y \mid \bar{A}_t = \bar{a}_t, \bar{X}_t = \bar{x}_t]$, but not $\mathbb{E}[Y(\bar{a}_t) \mid \bar{X}_t = \bar{x}_t]$, which are in general not equal. Moreover, we cannot test whether sequential strong ignorability holds in practice, since we never observe all potential outcomes. Instead, we only observed the outcome associated to the assigned treatment history, which is not sufficient to test the conditional independence. Figure 1 illustrates the general setup.

In this paper, we address this problem and show how to estimate individualized treatment effects over time in presence of unobserved confounders.

## 4 Sequential Deconfounder

We introduce the Sequential Deconfounder, a method that enables the estimation of treatment responses in presence of unobserved confounders when treatments are assigned sequentially, i.e., over time. The Sequential Deconfounder proceeds in two steps: (1) We deconfound the data. For this, we leverage the sequential dependence between assigned treatments to infer latent variables which act as substitutes for the unobserved confounders. (2) We estimate an outcome model using the observational data augmented with the substitutes. We prove that this yields unbiased estimates of individualized treatment responses over time.

### 4.1 Step 1: Deconfounding

Our approach for sequential deconfounding is based on the following idea. If there are unobserved confounders $U_t$, they introduce dependence between the treatments assigned over time. Hence, we can use the probabilistic law according to which the assigned treatments in the observed data changes over time to infer latent variables $Z_t \in \mathcal{Z}_t$. These latent variables then act as substitutes for the unobserved confounders. We prove that there cannot exist unobserved confounders that are not captured by $Z_t$ and, thus, conditioning on $Z_t$ yields an unbiased estimate of the individualized treatment response.

#### 4.1.1 Conditional Markov Model

We introduce the following conditional Markov model (CMM). The CMM determines the probabilistic laws according to which the assigned treatments in the observed data changes over time. Conditional on the observed covariates $X_t$ and the latent variable $Z_t$, the CMM renders the treatment assignment as a Markov process.

**Definition 1.** *(Conditional Markov model for sequential treatments.) The assigned treatments follows a conditional Markov model if the distribution can be written as*

$$p(\bar{a} \mid \bar{x}, \bar{z}, \theta) = p(a_1 \mid z_1, x_1, \theta) \prod_{t=2}^{T} p(a_t \mid a_{t-1}, z_t, x_t, \theta), \tag{2}$$

*where $\theta$ are parameters.*

Notice that we do not assume that, in the observational data, the patient covariates $x_t$ at timestep $t$ are independent of the patient history. We can leverage the sequential dependence between assigned treatments over time in the CMM to infer a latent variable $Z_t$ which, conditional on the covariates $X_t$, renders the treatment assignment Markovian.

Markovianity is realistic in practice, since clinicians typically determine the course of treatment considering not the entire treatment history, but the current treatment. As such, some of the most established methods for modeling decision-making in medicine are based on Markov processes (e. g., Bennett and Hauser, 2013; Komorowski et al., 2018; Steimle and Denton, 2017; Tsoukalas et al., 2015).

#### 4.1.2 Theoretical Results

In this section, we prove that, when the assigned treatments follow a CMM, the latent variables $Z_t$ act as valid substitutes for the unobserved confounders. In particular, we show that there cannot exist any other unobserved confounder $U_t$ that is not captured by $Z_t$. If there existed another unobserved confounder $U_t$, then the conditional Markov property of the treatment assignment would not be satisfied anymore.

This argument does not apply to time-varying unobserved confounders, i. e., unobserved confounders that change over time and affect one assigned treatment at each timestep. Hence, we require time-invariant unobserved confounding.[2]

**Assumption 3.** *(Time-Invariant Unobserved Confounding.) All unobserved confounders $U_t$ are time-invariant, i. e., $U_t = U$, for all $t \in \{1, \ldots, T\}$.*

Assumption 3 requires the unobserved confounder to be the same random variable at each timestep, but potentially different for each patient.

Time-invariant unobserved confounding is realistic in many domains. For instance, in the medical domain, social data such as socio-economic status or housing condition are, over the course of treatment, time-invariant. Although these variables are potential confounders for treatment decisions and explain potential disparities associated with the outcome of interest, social data is not routinely captured in electronic health records (EHRs) (Cantor and Thorpe, 2018; Chan et al., 2010). Moreover, social determinants of health that could be imported from data sources such as social services organizations are usually missing in EHRs due to the lack of interoperability (Adler and Stead, 2015). Our Sequential Deconfounder is suitable for capturing such unobserved confounders. As a direct consequence of Assumption 3, the latent variable $Z_t$ is also time-invariant.

We now present the connection between the CMM and sequential strong ignorability, which justifies the use of our method.

**Theorem 1.** *If the treatment assignment can be written as a CMM, we obtain sequential strong ignorability, conditional on the substitute and the covariates, i. e.*

$$Y(\bar{a}_t) \perp\!\!\!\perp A_t \mid \bar{A}_{t-1}, \bar{X}_t, Z, \tag{3}$$

---

[2]In epidemiology, such confounders are often called *missing baseline measurements* (White and Thompson, 2005), whereas they are called *time-invariant unobserved heterogeneity* in economics (Plümper and Troeger, 2007).

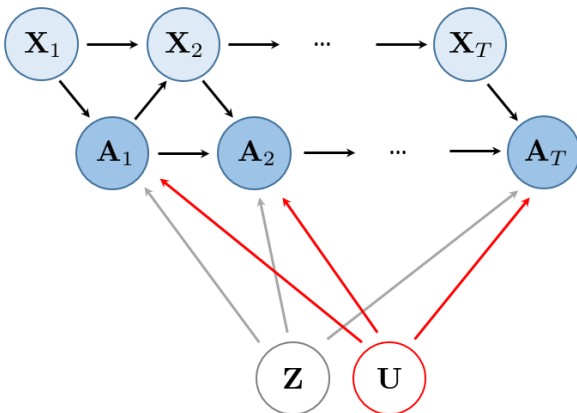

Figure 2: An illustration of the proof of Theorem 1. If there existed an unobserved time-invariant confounder $U$ that is not captured by $Z$, it would introduce dependence between the assigned treatments beyond the previous treatment (depicted by the red arrows), which contradicts the Markov property imposed by the CMM.

*for all $\bar{a}_t \in \bar{\mathcal{A}}_t$ and for all $t \in \{1, \ldots, T\}$.*

*Sketch of Proof.* The Markov process in the CMM requires that treatment $a_t$ only depends on the previous treatment $a_{t-1}$, given covariates $x_t$ and latent variable $z$. If there was a time-invariant unobserved confounder $u$ that is not captured by $z$, then this would introduce dependence between assigned treatments beyond the previous treatment $a_{t-1}$. This would contradict the Markov property of the treatment assignment. See Appendix A for a full proof. □

Theorem 1 has several implications. First, using the latent variable $Z$ inferred by the CMM as a substitute for the unobserved confounder yields unbiased estimates of individualized treatment responses. Second, it shows how to find a valid substitute for the unobserved confounder that renders the treatments subject to sequential strong ignorability. A valid substitute has to satisfy two sufficient conditions: (i) the substitute comes from a CMM; (ii) the CMM captures the distribution of the assigned treatments, which both can be tested. In sum, Theorem 1 confirms that the Sequential Deconfounder enables unbiased estimation of individualized treatment responses over time when the treatment assignment can be written as a CMM.[3]

In order to assess the quality of the fitted CMM in the Sequential Deconfounder, we can use predictive model checks as introduced in previous work (e. g., Bica et al., 2020b; Rubin, 1984). We describe predictive model checks in Appendix C.

## 4.2 Step 2: Outcome Model

Once we obtained an estimate $\hat{Z}$ for the substitute, we use it to estimate $\mathbb{E}\left[Y \mid \bar{A}_t = \bar{a}_t, \bar{X}_t = \bar{x}_t, Z = \hat{z}\right]$ via an outcome model (e. g., Bica et al., 2020a; Lim et al., 2018; Robins and Hernán, 2009). While any outcome model can be used, we use established outcome models for estimating individualized treatment responses over time as discussed in Section 6.2.

## 4.3 "Time-Invariant Unobserved Confounders" vs. "No Unobserved Confounders"

We compare the assumption of the Sequential Deconfounder to the "no unobserved confounders" assumption. Most existing methods for estimating treatment responses over time assume that there are no unobserved confounders at all (e. g., Lim et al., 2018; Robins, 1986). In practice, however, this assumption rarely holds, since many confounders are not subject to data collection or cannot be easily measured. Our work makes *strictly weaker* assumptions: We assume that there are only time-invariant unobserved confounders. This

---

[3]The static deconfounder framework introduced by Wang and Blei (2019a) initiated several discussions regarding identification of the latent variable model (e. g., D'Amour, 2019), which we discuss for our setting in detail in Appendix B.

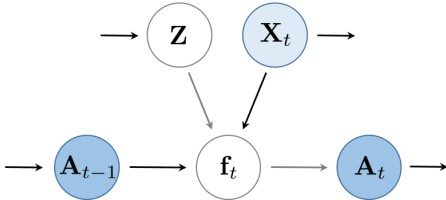

Figure 3: An Illustration of the SeqGPLVM. The current treatment $A_t$ is determined by the covariates $X_t$, the latent variable $Z$, and the previous treatment $A_{t-1}$. The function $f_t$ is modeled by a Gaussian process.

is relevant in practice. For instance, social data such as socio-economic status or housing conditions are, over the course of treatment, time-invariant. Yet, besides being confounders, they are not recorded in EHRs (Chan et al., 2010).

## 5  Instantiation of the Sequential Deconfounder

In this section, we propose an instantiation of the Sequential Deconfounder in order to infer the substitute $Z$. Note that our theory holds true for any model that infers the latent variable $Z$ in the CMM: it does not restrict the class of models that can be used. However, factor models (Bica et al., 2020b; Wang and Blei, 2019a) are prohibited in the sequential setting, since they render the assigned treatments independent of each other and, hence, cannot capture any sequential dependence between the assigned treatments. As a remedy, we develop a sequential Gaussian process latent variable model (SeqGPLVM) that captures the sequential dependencies between assigned treatment by leveraging the underlying structure of the CMM.

### 5.1  The SeqGPLVM

In the context of latent variable models, the Gaussian process latent variable model (GPLVM) is a prominent example for probabilistic learning of latent variables (e.g., Lawrence, 2004; Titsias and Lawrence, 2010). However, standard GPLVMs are typically not applicable to the Sequential Deconfounder, since they were developed for the static setting and, hence, cannot capture sequential dependencies between treatment assignment over time. Tailored GPLVMs were proposed that allow some of the covariates to be observed and others to be latent (Märtens et al., 2019), but again in the static setting. As a remedy, we introduce a novel SeqGPLVM in the following.

Given the covariates and treatment history, $\bar{x}_t$ and $\bar{a}_t$, our objective is to infer the posterior distribution of the substitute, $p(z \mid \bar{a}_t, \bar{x}_t)$, over latent coordinates whilst maintaining the latent structure imposed by the CMM. To this end, we place a prior over the substitute, $p(Z) = \mathcal{N}(z \mid 0, \sigma^2 I)$ and define the forward mappings $f_t : \mathcal{Z} \times \mathcal{X}_t \times \mathcal{A}_{t-1} \to \mathcal{A}_t$, i.e., the mapping from the joint space representing the substitute, the observed covariates, and the previous treatment assignment to the space of the current treatment assignment. We choose Gaussian processes (GPs) as a non-parametric model for the forward mappings, i.e., $f_t \sim \mathcal{GP}(0, K)$. This choice reflects the strong theoretical underpinnings of GPs and advances that enabled such models to be scalable to large datasets (Hensman et al., 2013). As a result, we obtain the following generative model, which maintains the structure of the CMM:

$$p(\bar{a}_t \mid \bar{f}_t, \bar{x}_t, z, \theta) = p(a_1 \mid f_1)\, p(f_1 \mid x_1, z, \theta) \prod_{s=2}^{t} p(a_s \mid f_s)\, p(f_s \mid a_{s-1}, x_s, z, \theta), \tag{4}$$

where $f_s \sim \mathcal{GP}(0, K)$. The generative model is depicted in Figure 3. Different forms of interactions between $z$, $x_t$, and $a_{t-1}$ can be encoded via different kernels. We use a treatment-adjusted kernel on the joint space $\mathcal{Z} \times \mathcal{X}_t \times \mathcal{A}_{t-1}$, which is based on the popular squared exponential kernel:

$$k^{\text{treat}}((z,x,a),(z',x',a')) = -\frac{1}{2}\sigma_{zxa}^2 \left[ \sum_{j=1}^{|\mathcal{Z}|} \left( \frac{z_j - z_j'}{l_j^{(z)}} \right)^2 + \sum_{j=1}^{|\mathcal{X}_t|} \left( \frac{x_j - x_j'}{l_j^{(x)}} \right)^2 + \sum_{j=1}^{|\mathcal{A}_{t-1}|} \left( \frac{a_j - a_j'}{l_j^{(a)}} \right)^2 \right], \tag{5}$$

where $\sigma_{zxa}^2$ is the kernel variance parameter and $l_j$ are the lengthscales on the different spaces. This kernel allows to capture interactions between $x_t$ and $z_t$ while adjusting for the previously assigned treatment $a_{t-1}$, but not for any assigned treatments further in the past as required by the CMM. Finally, we make use of Bayesian shrinkage priors on the kernel variance $\sigma_{zxa}^2$ to encourage unnecessary components to be shrunk to zero, i.e., we specify $\sigma_{zxa}^2 \sim \Gamma(1,1)$.

## 5.2 Inference for SeqGPLVM

To make inference, we adopt variational inducing point based inference (Titsias and Lawrence, 2010) to the SeqGPLVM in a straightforward manner. Recall that the treatment-adjusted kernel is defined on the product space $\mathcal{Z} \times \mathcal{X}_t \times \mathcal{A}_{t-1}$, and hence the inducing points lie in this space which has dimensionality $|\mathcal{Z}| + |\mathcal{X}_t| + |\mathcal{A}_{t-1}|$. The emission likelihood $p(a_t \mid f_t)$ determines the marginal likelihood and has to be chosen depending on the space $\mathcal{A}_t$. For instance, if $\mathcal{A}_t = \{0,1\}$, one might choose a Bernoulli likelihood. If treatments are continuous, the emission likelihood could be Gaussian. In the latter case, one can analytically integrate out the GP mappings and give a closed-form solution for the marginal likelihood.

## 5.3 Advantages of our SeqGPLVM

Our theory holds true for any model that infers the latent variable $Z$ in the CMM: it does not restrict the class of models that can be used. However, off-the-shelf latent variable models that allow some confounders to be observed and others to be unobserved are scarce, in particular, in the sequential setting. Moreover, there are several advantages of using SeqGPLVM: (i) The Bayesian approach allows to infer the posterior distribution of the substitute $Z$. This allows to quantify the uncertainty originating from the Sequential Deconfounder in the estimate of the individualized treatment responses. This can be done as follows: we first draw $s$ samples $\{z^{(1)}, \ldots, z^{(s)}\}$ of the substitute from the posterior distribution, i.e., $z^{(l)} \sim p(z \mid \bar{a}_t, \bar{x}_t)$. For each sample $z^{(l)}$, we then fit an outcome model and compute a point estimate of the individualized treatment response. We aggregate the estimates of the treatment responses from the $s$ samples. The variance of these aggregated estimates describes the uncertainty of the Sequential Deconfounder. (ii) It enables us to easily sample from the marginal likelihood to obtain samples of treatments. These samples are used to compute predictive model checks over time, which are necessary to assess the model fit of the SeqGPLVM (cf. Rubin, 1984). (iii) The use of GPs enables us to learn complex and nonlinear relationships from the data, which is needed in medical application involving complex diseases.

# 6 Experiments on Simulated Data

The medical domain is known to be prone to many unobserved confounders affecting both treatment assignment and outcomes (Gottesman et al., 2019). As such, estimating individualized treatment responses requires adaptation to the sequential setting with unobserved confounders. We simulate a medical environment which offers the advantage that we have access to ground truth individualized treatment responses. Further, we can vary the amount of unobserved confounding in order to validate our theory empirically similar to Bica et al. (2020b); Louizos et al. (2017); Wang and Blei (2019a).

## 6.1 Data Simulation

We simulate observational data $\mathcal{D} = (\{x_t^{(i)}, a_t^{(i)}, y_{t+1}^{(i)}\}_{t=1}^{T^{(i)}})_{i=1}^N$ from a medical environment similar to previous works (Bica et al., 2020b). This environment consists of a patient's (observed) medical state $X_t$ at each timestep, which changes according to the treatment history and past medical states by

$$X_{t,j} = \frac{1}{p} \sum_{i=1}^{p} (\alpha_{i,j} X_{t-i,j} + \omega_i A_{t-i}) + \eta_t, \tag{6}$$

with weights $\alpha_{i,j} \sim \mathcal{N}(0, 0.5^2)$, $\omega_i \sim \mathcal{N}(1 - (i/p), (1/p)^2)$ and noise $\eta_t \sim \mathcal{N}(0, 0.01^2)$. The initial medical state is given by $X_{0,j} \sim \mathcal{N}(0, 0.1^2)$. Suppose there exists a time-invariant unobserved confounder $U$, which is modeled by

$$U \sim \mathcal{N}(0, 0.1^2). \tag{7}$$

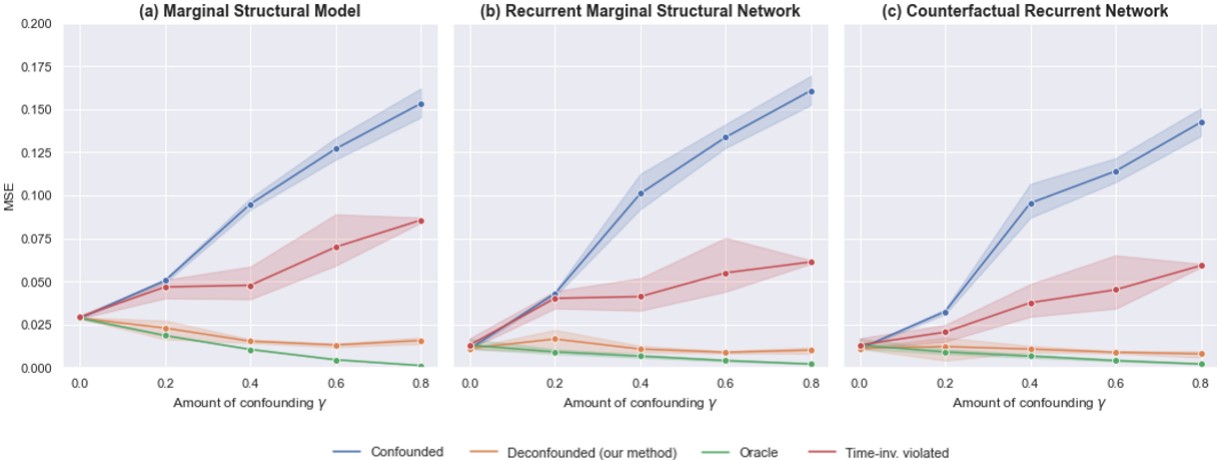

Figure 4: Mean squared error (MSE) and standard deviation for one-step ahead prediction of treatment responses for varying amount of unobserved confounding. We use (a) Marginal Structural models, (b) Recurrent Marginal Structural Networks, and (c) Counterfactual Recurrent Networks as outcome models. Lower is better.

This confounder is not included in the data and, thus, is an unobserved confounder. The treatment assignments depends on the medical state $X_t$, the unobserved confounder $U$, and the previous treatment $A_{t-1}$:

$$\pi_t = \gamma_A U + (1 - \gamma_A)(\hat{X}_t + A_{t-1}), \tag{8}$$

$$A_t \mid \pi_t \sim \text{Bernoulli}(\sigma(\lambda \pi_t)), \tag{9}$$

where $\hat{X}_t$ is the mean of the covariates, $\lambda = 15$, $\sigma(\cdot)$ is the sigmoid function, and $\gamma_A \in [0, 1]$ is a parameter which controls the amount of confounding applied to the treatment assignment. No unobserved confounding corresponds to $\gamma_A = 0$. The outcome is a function of unobserved confounder and medical state:

$$Y_{t+1} = \gamma_Y U + (1 - \gamma_Y)\hat{X}_{t+1}, \tag{10}$$

where, as before, $\gamma_Y \in [0, 1]$ controls the amount of unobserved confounding applied to the outcome. We simulate 5,000 patients trajectories from the medical environment with 20 to 30 timesteps. The dimension and time-dependence of the covariates is set to $k = 3$ and $p = 3$. Each dataset is split 80/20 for training and testing, respectively.

## 6.2  Outcome Model

We evaluate the effectiveness of our Sequential Deconfounder to adjust for unobserved confounders when used *in conjunction* with an outcome model for estimating treatment responses. We focus on three established outcome model: **(a)** Marginal Structural Models (e. g., Robins and Hernán, 2009; Robins et al., 2000) from the epidemiology literature, **(b)** Recurrent Marginal Structural Networks (Lim et al., 2018), and **(c)** Counterfactual Recurrent Networks (Bica et al., 2020a) from the machine learning literature. Note that any outcome model can be used in conjunction with our Sequential Deconfounder.

## 6.3  Results

In this section, we use the Sequential Deconfounder to obtain unbiased estimates of one-step ahead treatment responses. For a comparative evaluation, we investigate three different scenarios in which the outcome models are trained to estimate treatment responses: **(i)** without information about the unobserved confounder $U$ ("Confounded"), **(ii)** with the true unobserved confounder $U$ ("Oracle"), and **(iii)** with the substitute $\hat{Z}$ obtained by the Sequential Deconfounder ("Deconfounded"). Note that we cannot compare to any other deconfounder methods such as Bica et al. (2020b), since our work is the first deconfounder method applicable in a setting in which there is only a *single* treatment available at each timestep. Hence, methods such as the

one of Bica et al. (2020b), which require multiple treatments assigned at each timestep in order for both their theory and factor model to work, are not applicable.

In Figure 4, we present the mean squared error (MSE) for one-step ahead estimation of individualized treatment responses for varying amount of unobserved confounding (i. e., varying the parameter $\gamma = \gamma_A = \gamma_Y$). We find that the Sequential Deconfounder, when used in conjunction with outcome models, improves the MSE substantially. This shows the strong benefit of our Sequential Deconfounder. In particular, the Sequential Deconfounder achieves unbiased estimates of treatment responses, i. e., its estimates are close to the estimates obtained by the oracle approach (which has access to the true unobserved confounder). In absence of unobserved confounding (i. e., $\gamma = 0$), using the Sequential Deconfounder does not decrease the performance.

**Violation of "Time-Invariant Unobserved Confounders".** We investigate the sensitivity of the Sequential Deconfounder with respect to violation of Assumption 3, i. e., when there are not only time-invariant unobserved confounders. For this, remove the time-varying covariate $X_{t,1}$ from the dataset. This makes $X_{t,1}$ a time-varying unobserved confounder, which violates Assumption 3. We then train the Sequential Deconfounder. We denote this scenario as "Time-inv. violated". In Figure 4, we observe that this yields biased estimates of the treatment responses. However, the performance of the Sequential Deconfounder remains superior to the performance when there is no control over unobserved confounders at all.

**Predictive checks.** Similar to previous works, we assess whether the Sequential Deconfounder captures the distribution of assigned treatments by performing predictive model checks (Bica et al., 2020b; Wang and Blei, 2019a). We observe that the $p$-values are close to the optimal value of 0.5, which means that our method captures the distribution of the assigned treatment. Details are given in Appendix D.

In presence of unobserved confounders, methods that rely on the assumption of no unobserved confounders yield biased estimates of the individualized treatment response, since $\mathbb{E}[Y(\bar{a}_t) \mid \bar{X}_t = \bar{x}_t] \neq \mathbb{E}[Y \mid \bar{A}_t = \bar{a}_t, \bar{X}_t = \bar{x}_t]$. In contrast, by inferring a substitute and augmenting the data with it, the Sequential Deconfounder provides unbiased estimates of individualized treatment responses. As a result, our Sequential Deconfounder, when used in conjunction with state-of-the-art methods, improves performance substantially.

## 7 Experiments on Real-World Data

We apply the Sequential Deconfounder to a real-world medical setting using the Medical Information Mart for Intensive Care (MIMIC-III) database (Johnson et al., 2016). The database consists of EHRs from patients in the intensive care unit. We extract $3,487$ patients with trajectories up to 30 timesteps and 25 patient covariates such as vital signals and lab tests together with static covariates such as gender. We estimate the individualized treatment response on two datasets. In particular, we estimate the individualized treatment response of: (1) vassopressors and (2) mechanical ventilators over time on three patient outcomes: white blood cell count, blood pressure, and oxygen saturation.

Table 1: Results (MSE) for estimating responses to *vasopressors* on MIMIC-III. Each outcome model is trained without information about the unobserved confounder $U$ (Conf.) and with the substitute $\hat{Z}$ obtained by Sequential Deconfounder (Deconf.) for 10 runs. Lower is better.

| | | MIMIC-III (Mean $\pm$ Std) | | |
|---|---|---|---|---|
| | | WHITE BLOOD CELL COUNT | BLOOD PRESSURE | OXYGEN SATURATION |
| **MSM** | CONF. | $.871 \pm .00$ | $.171 \pm .00$ | $.497 \pm .00$ |
| | DECONF. | $\mathbf{.709 \pm .01}$ | $\mathbf{.165 \pm .00}$ | $\mathbf{.337 \pm .01}$ |
| **RMSN** | CONF. | $.677 \pm .03$ | $.135 \pm .01$ | $.401 \pm .02$ |
| | DECONF. | $\mathbf{.532 \pm .04}$ | $\mathbf{.101 \pm .01}$ | $\mathbf{.249 \pm .02}$ |
| **CRN** | CONF. | $.665 \pm .05$ | $.081 \pm .00$ | $.357 \pm .02$ |
| | DECONF. | $\mathbf{.521 \pm .06}$ | $\mathbf{.073 \pm .00}$ | $\mathbf{.199 \pm .03}$ |

We use the Sequential Deconfounder in conjunction with the same outcome models as in Section 6. In Table 1, we illustrate the MSE for one-step ahead estimation of vassopressors. The results for mechanical ventilators can be found in Appendix E. Note that we do not have access to the true treatment responses, since MIMIC-III is a real-world medical dataset. However, compared to the standalone outcome models ("Conf."), using our Sequential Deconfounder and augmenting the data with the substitutes for the unobserved confounders ("Deconf.") yields substantially lower MSE for the response to vassopressors. This is because the Sequential Deconfounder uses the sequential dependence between assigned treatments to infer latent variables which account for unobserved variables. By using these latent variables as substitutes, we improve the estimation of individualized treatment responses. While these results on real-world data require validation from clinicians, they indicate the potential benefit of the Sequential Deconfounder in real medical settings.

## 8   Conclusion

We propose the Sequential Deconfounder, which exploits the treatment assignment over time to infer substitutes for unobserved confounders. These substitutes are used in conjunction with an outcome model to estimate treatment responses over time. On simulated and real data, we show our method's benefit for estimating treatment responses with unobserved confounders. A directions for future research is the Markovianity of the CMM. Here, the current treatment assignment is only dependent on the last treatment assignment, but not on any treatment assignments prior to this. Many established methods for modeling medical decision-making are based on Markovianity, which may be realistic in many applications. However, it may be restrictive in settings in which Markovianity does not hold true. As such, extending upon the CMM is an interesting avenue for future research.

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

## A  Proof of Theorem 1

In order to prove Theorem 1, we define the *recursive construction* and prove lemmas that are used for the proof of Theorem 1. As a reminder, in order to obtain sequential strong ignorablility using the substitute $Z$ for the unobserved confounder, the following needs to hold:

$$Y(\bar{a}_t) \perp\!\!\!\perp A_t \mid \bar{A}_{t-1}, \bar{X}_t, Z, \tag{11}$$

for all $\bar{a}_t \in \bar{\mathcal{A}}_t$ and for all $t = 1, \ldots, T$.

**Definition 2.** *(Recursive construction.) The sequence of assigned treatments $(A_t)_{t \geqslant 1}$ admits a recursive construction from $Z$ and $(X_t)_{t \geqslant 1}$, if at any timestep $t$, there exist a (deterministic) measurable function $f_t : \mathcal{Z} \times \mathcal{X}_t \times \mathcal{A}_{t-1} \times [0,1] \to \mathcal{A}_t$ and random variables $V_t$, such that the distribution of the assigned treatment $A_t$ can be written as*

$$A_t = f_t(A_{t-1}, Z, X_t, V_t), \tag{12}$$

*where $V_t \sim Uniform([0,1])$ and satisfies*

$$V_t \perp\!\!\!\perp Y(\bar{a}_t) \mid \bar{A}_{t-1}, \bar{X}_t, Z \tag{13}$$

*for all $\bar{a}_t \in \bar{\mathcal{A}}_t$.*

**Lemma 1.** *(Recursive construction is sufficient for sequential strong ignorability.) If the sequence of assigned treatments $(A_t)_{t \geqslant 1}$ admits a recursive construction from $Z$ and $(X_t)_{t \geqslant 1}$ then we obtain sequential strong ignorability.*

*Proof.* Without loss of generality it is assumed that $\mathcal{A}_t$ is a Borel space and $\mathcal{Z}$ and $\mathcal{X}_t$ are measurable spaces for any $t \in \{1, \ldots, T\}$. Because $(A_t)_{t \geqslant 1}$ admits a recursive construction from $Z$ and $(X_t)_{t \geqslant 1}$, we can write $A_t = f_t(A_{t-1}, Z, X_t, V_t)$, where $f_t$ is measurable and

$$V_t \perp\!\!\!\perp Y(\bar{a}_t) \mid \bar{A}_{t-1}, \bar{X}_t, Z \tag{14}$$

for all $\bar{a}_t \in \bar{\mathcal{A}}_t$. This implies that

$$(A_{t-1}, Z, X_t, V_t) \perp\!\!\!\perp Y(\bar{a}_t) \mid \bar{A}_{t-1}, \bar{X}_t, Z. \tag{15}$$

Since $A_t$ is a measurable function of $(A_{t-1}, Z, X_t, V_t)$, sequential strong ignorability holds true, i.e.,

$$A_t \perp\!\!\!\perp Y(\bar{a}_t) \mid \bar{A}_{t-1}, \bar{X}_t, Z, \tag{16}$$

for all $\bar{a}_t \in \bar{\mathcal{A}}_t$ and for all $t = 1, \ldots, T$. □

**Lemma 2.** *(Conditional Markov model for sequential treatments is sufficient for recursive construction.) Under weak regularity conditions, if the distribution of the assigned treatments $p(\bar{a})$ can be written as a conditional Markov model, then the sequence of assigned treatments obtains a recursive construction.*

Regularity condition: the set of treatments $\mathcal{A}_t$ is a Borel subset of compact intervals. Without loss of generality, it is assumed that $\mathcal{A}_t = [0,1]$ for all $t = 1, \ldots, T$.

The proof for Lemma 2 uses Proposition 7.6 in Kallenberg (2006) (recursion): Let $(X_t)_{t \geqslant 1}$ be a sequence of random variables with values in a Borel space $S$. Then $(X_t)_{t \geqslant 1}$ is Markov if and only if there exist some measurable functions $f_t : S \times [0,1] \to S$ and random variables $V_t \overset{iid}{\sim} Uniform([0,1])$ with $V_t \perp\!\!\!\perp X_1$ such that $X_t = f_t(X_{t-1}, V_t)$, $\mathbb{P}$-a.s., for all $t \geqslant 1$. $(X_t)_{t \geqslant 1}$ is time-homogeneous if and only if $f_1 = f_2 = \ldots = f$.

*Proof.* At timestep $t$, consider the random variables $A_t \in \mathcal{A}_t$, $Z \in \mathcal{Z}$, and $X_t \in \mathcal{X}_t$. Since the distribution of the assigned treatments can be written as a conditional Markov model, we know that the sequence of assigned treatments, when conditioned on $Z$ and $X_t$, i.e., $(A_t \mid X_t, Z)_t$, is Markov. Hence, from Proposition 7.6 in Kallenberg (2006), there exits some measurable function $f_t : \mathcal{Z} \times \mathcal{A}_{t-1} \times \mathcal{X}_t \times [0,1] \to \mathcal{A}_t$ such that

$$A_t = f_t(A_{t-1}, Z, X_t, V_t), \tag{17}$$

with $V_t \overset{iid}{\sim} \text{Uniform}([0,1])$ and $V_t \perp\!\!\!\perp A_1$ for all $t = 1, \ldots, T$. It remains to show that

$$V_t \perp\!\!\!\perp Y(\bar{a}_t) \mid \bar{A}_{t-1}, \bar{X}_t, Z. \tag{18}$$

This can be seen by distinction of cases. If there exists a random variable $U_t$ (not equal to $Z$ or $X_t$ almost surely) that confounds $V_t$ and $Y(\bar{a}_t)$, it is either (i) time-invariant or (ii) time-varying. (i) If $U_t$ is time-invariant, then it would also confound $V_s$ for $s \neq t$, which introduces dependence between the random variables $V_t$ for $t = 1, \ldots, T$. However, since $V_t$ are drawn *iid* from $\text{Uniform}([0,1])$, this cannot be the case. Otherwise, $V_t$ and $V_s$ for $s \neq t$ would not be jointly independent. (ii) If $U_t$ is time-varying, then $U_t$ would confound $A_t$ through $V_t$. As a consequence, $U_t$ would be also a confounder for $A_t$. However, because of Assumption 3, there cannot be any time-varying confounders for $A_t$. As a result, there cannot be another random variable that confounds $V_t$, and therefore $V_t \perp\!\!\!\perp Y(\bar{a}_t) \mid \bar{A}_{t-1}, \bar{X}_t, Z$ holds true. $\qquad \square$

## B  Identifiability of the Individualized Treatment Response

The deconfounder framework introduced by Wang and Blei (2019a) in the static setting initiated several independent discussion regarding new theory and applications (Athey et al., 2019; D'Amour, 2019; Imai and Jiang, 2019; Ogburn et al., 2019; Wang and Blei, 2019b). In particular, issues regarding the identification[4] of the causal quantity $\mathbb{P}(Y \mid do(A = a))$ were raised in the static setting with multiple treatments (D'Amour, 2019). This issue arises when the substitute confounder cannot be uniquely recovered from the observational data. However, under the conditions in Wang and Blei (2019a), the identification of the mean potential outcome is indeed given. This relies particularly on the condition of consistent substitute confounders,[5] which allows to uniquely recover the substitute confounder (see Theorem 8 in Wang and Blei (2019a) and explanations in Wang and Blei (2019b; 2020)).

This discussion also applies to our work. Indeed, if the substitute confounder is not uniquely determined by the Markov model, then we cannot be sure that we recover the correct one. However, the identifiability is given under similar conditions on the consistency of the substitute confounder as in Wang and Blei (2019a). This ensures that we can pinpoint the substitute confounder and then use it in the outcome model to estimate the individualized treatment response. The proof in our setting is similar to the proof in the static setting with multiple treatments (Theorem 8 in Wang and Blei (2019a)). For completeness, we give a full proof in our setting below. Moreover, the identifiability of the Sequential Deconfounder is also empirically supported by the experimental results in Section 6, in which the Sequential Deconfounder achieves oracle-near estimation errors.

Before stating the identification results, we first describe the notion of a consistent substitute confounder; we will rely on this notion for identification.

**Definition 3.** *(Consistency of the substitute confounder.) The CMM admits consistent estimates of the substitute confounder $Z$ if, for some function $f$,*

$$p(z \mid \bar{x}, \bar{a}) = \delta_{f(\bar{x}, \bar{a})}. \tag{19}$$

Consistency of substitute confounders yields that we can estimate the substitute confounder $Z$ from the treatment history $\bar{a}$ and the covariate history $\bar{x}$ with certainty. Since it is a deterministic function of the treatment and covariate history, it uniquely pinpoints the substitute confounder. Nevertheless, the substitute confounder need not coincide with the true data-generating $Z$, nor does it need to coincide with the true unobserved confounder. We only need to estimate the substitute confounder $Z$ up to some deterministic bijective transformations (e. g., scaling and linear transformations).

**Theorem 2.** *(Identifiability of the individualized treatment response.) Assume time-invariant confounding and consistent substitute confounders. Then, for the assigned treatment history $\bar{a}$, the individualized treatment response to an alternative assignment $\bar{a}'$ is*

$$\mathbb{E}[Y(\bar{a}') \mid Z = z, \bar{X} = \bar{x}] = \mathbb{E}[Y \mid Z = z, \bar{X} = \bar{x}, \bar{A} = \bar{a}']. \tag{20}$$

---

[4]In this context, identifiability means that a quantity can be written as a function of the observed data.

[5]The substitute confounder is consistent, if it is determined by a deterministic function of the treatment assignment (Wang and Blei, 2019a).

*This holds when the alternative assignment $\bar{a}'$ leads to the same substitute confounder estimate as the observed treatment history, i. e., $f(\bar{a}, \bar{x}) = f(\bar{a}', \bar{x})$.*

*Proof.* To prove identification, we rewrite the individualized treatment response

$$\mathbb{E}[Y(\bar{a}') \mid Z = z, \bar{X} = \bar{x}] \tag{21}$$
$$= \mathbb{E}[Y(\bar{a}') \mid Z = z, \bar{X} = \bar{x}, \bar{A} = \bar{a}] \tag{22}$$
$$= \mathbb{E}[Y(\bar{a}') \mid Z = f(\bar{a}, \bar{x}), \bar{X} = \bar{x}, \bar{A} = \bar{a}] \tag{23}$$
$$= \mathbb{E}[Y(\bar{a}') \mid Z = f(\bar{a}, \bar{x}), \bar{X} = \bar{x}, \bar{A} = \bar{a}'] \tag{24}$$
$$= \mathbb{E}[Y \mid Z = f(\bar{a}, \bar{x}), \bar{X} = \bar{x}, \bar{A} = \bar{a}'] \tag{25}$$
$$\tag{26}$$

The first equality makes use of the unconfoundedness given $Z$, and $\bar{X}$ by Theorem 1. The second equality makes use of the consistency of the substitute confounder. The third equality makes again use of the unconfoundedness and $f(\bar{a}, \bar{x}) = f(\bar{a}', \bar{x})$. The fourth equation is estimable from data, since $f(\bar{a}, \bar{x}) = f(\bar{a}', \bar{x})$. Hence, the identification of $\mathbb{E}[Y(\bar{a}') \mid Z = z, \bar{X} = \bar{x}]$ is provided. □

The condition $f(\bar{a}, \bar{x}) = f(\bar{a}', \bar{x})$ requires that the changing the assignment does not change the substitute confounder. This is particularly reasonable in the case of time-invariant confounder, since such confounder do no change over time. For instance, considering an alternative treatment assignment or half way though the therapy changing the treatment assignment would not change the housing condition of the patient.

## C  Predictive Checks

We provide a sound definition of predictive model checks over time (e. g., Bica et al., 2020b; Rubin, 1984) in order to assess the quality of the fitted CMM in the Sequential Deconfounder. We do this since the CMM has to appropriately capture the distribution of the treatment assignment. Hence, the quality of the causal estimate relies on the quality of the fitted CMM.

To this end, a CMM is fitted on a training dataset. Then, for each patient in a validation dataset, we sample $M$ treatment assignments $a_{t,rep}$ from the fitted CMM at each timestep. We compare the samples from the fitted CMM to the actual treatment assignment $a_{t,val}$ via the predictive $p$-value, which is computed as follows:

$$\frac{1}{M} \sum_{i=1}^{M} \mathbf{1}(T(a_{t,rep}^{(i)}) < T(a_{t,val})), \tag{27}$$

where $\mathbf{1}(\cdot)$ is the indicator function and $T(a_t)$ is the test statistic defined as

$$T(a_t) = \mathbb{E}_Z[\log p(a_t \mid X_t, Z)]. \tag{28}$$

The test statistics for the treatment samples $a_{t,rep}$ are similar to the test statistics for the treatments in the validation set, if the model captures the distribution of the assigned treatments well. In this case, the ideal $p$-value is 0.5.

## D  Predictive Model Checks for Experiments on Simulated Data

The quality of the treatment response estimates based on the Sequential Deconfounder relies on how well it captures the distribution of the assigned treatments, since it infers the substitute. Hence, we assess whether the Sequential Deconfounder captures the distribution of the assigned treatments for the experiments on simulated data in Section 6 by performing predictive model checks over time as in Bica et al. (2020b). The results are shown in Figure 5. We see that the $p$-values are close to the optimal value of 0.5. Hence, we are confident that the SeqGPLVM captures the distribution of the treatment assignment well.

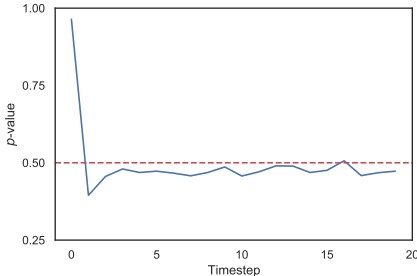

Figure 5: Predictive checks for SeqGPLVM to assess its quality of capturing the treatment assignment distribution. The red line indicates the optimal $p$-values of 0.5 as in Bica et al. (2020b). The blue line indicates the $p$-values of SeqGPLVM. We see that SeqGPLVM captures the treatment assignment distribution well as its $p$-values are close to 0.5.

Table 2: Results (MSE) for estimating responses to antibiotics on MIMIC-III. Each outcome model is trained without information about the unobserved confounder $U$ (Conf.) and with the substitute $\hat{Z}$ obtained by Sequential Deconfounder (Deconf.) for 10 runs. Lower is better. Best in bold.

**Dataset 2:** MIMIC-III w/ treatment: Mechanical Ventilator

| | | MIMIC-III (Mean $\pm$ Std) | | |
|---|---|---|---|---|
| | | White blood cell count | Blood pressure | Oxygen saturation |
| **MSM** | Conf. | $1.295 \pm .00$ | $2.418 \pm .00$ | $1.013 \pm .00$ |
| | Deconf. | $\mathbf{1.182 \pm .04}$ | $\mathbf{2.147 \pm .03}$ | $\mathbf{0.897 \pm .04}$ |
| **RMSN** | Conf. | $1.147 \pm .05$ | $2.173 \pm .03$ | $0.886 \pm .05$ |
| | Deconf. | $\mathbf{1.007 \pm .06}$ | $\mathbf{1.854 \pm .05}$ | $\mathbf{0.677 \pm .05}$ |
| **CRN** | Conf. | $1.112 \pm .07$ | $1.970 \pm .06$ | $0.807 \pm .05$ |
| | Deconf. | $\mathbf{0.937 \pm .08}$ | $\mathbf{1.694 \pm .08}$ | $\mathbf{0.593 \pm .06}$ |

## E  Additional Results on Real-World Data

Here, we present the results for estimating the individualized treatment response mechanical ventilators over time on three patient outcomes: white blood cell count, blood pressure, and oxygen saturation. Our Sequential Deconfounder in conjunction with the same outcome models as in Section 6 and Section 7. The results can be found in Table 2, where , we illustrate the MSE for one-step ahead estimation of mechanical ventilators. Compared to the standalone outcome models ("Conf."), using our Sequential Deconfounder and augmenting the data with the substitutes for the unobserved confounders ("Deconf.") yields substantially lower MSE for the response to mechanical ventilators.

