# OpenReview forum: "Sequential Deconfounding for Causal Inference with Unobserved Confounders"
_TMLR — Rejected by TMLR_

### Review · Reviewer_hN8v · 2022-07-24

**Summary Of Contributions:**

This paper proposes a theory of sequential deconfounding for individual treatment effect estimation when a single treatment is assigned sequentially over time. An instantiation of the theory is given with a sequential Gaussian process latent variable. Experiments demonstrate the improvement in applying the theory to practice.

**Requested Changes:**

I hope the authors can address the weaknesses mentioned above, especially 1,2,5,6,7.

**Strengths And Weaknesses:**

Strengths:
The paper studies the single treatment problem, which is currently missing in the sequential deconfounding literature. The theory and models have demonstrated strong empirical performance. The paper is easy to follow as a whole.

Weaknesses:
1. Presentation. The time-invariant assumption is not clear in the abstract and introduction. It misguides the readers to think  that this is a direct extension of the previous sequential deconfounding works with multiple treatments to a single treatment. Because the unmeasured confounder is time-invariant, i.e., static, the treatment effect estimation story has two parts: the observed part is sequential while the unobserved part is static. The setup is fundamentally different from the previous deconfounding works with multiple treatments.
2. Motivation. It is unclear why the existing theory with multiple treatments is not applicable to the setting with a single treatment, and why the assumption made in the paper is crucial to make this possible.
3. In Theorem 1, the random variable Z is not defined beforehand, is it a proxy of U under Assumption 3?
4. CMM. The limitations of the conditional Markov model are not discussed. It doesn't look like allowing the treatment assignment at time t to base on the entire history of covariates and treatment assignments.
5. Related works. Under the time-varying assumption, more discussion about related deconfounding works in the static setting is needed to explain why they are not applicable to the proposed problem (with the help of some time-series models to model the observed outcomes and assignments over time)
6.  The predictive model check inherited from (e. g., Bica et al., 2020b; Rubin, 1984) has no theoretical guarantee, so eventually, we don't know it the deconfounding is successful or not.
7. The variational inducing point-based inference is a scalable method for GP regression. As the sample size increases, the method would still only use a subset of samples to approximate the exact GP solution.  If the unobserved confounder has a complicated distribution then its learning should take a large number of samples. So the inducing point method may never converge to the truth.
8. The authors mentions serval times in the paper the unbiased estimation of individualized treatment effects (ITEs). The ITE is a conditional mean. In nonparametric regression, even we have all the observed covariates X, any regression model from X to Y is going to be biased in finite samples. I think the word "unbiased" is misused in the paper or it should be defined mathematically.

---

> ### Author Response · Authors · 2022-08-07
> **Response to Reviewer hN8v**
>
> We thank Reviewer hN8v for the valuable feedback and comments. Below we respond in detail to each of the points. Please also see the revised version of our paper.
>
> 1. Our work addresses sequential deconfounding in the case of a single treatment (instead of multiple treatments) at each timestep. As such, in comparison to the works on sequential deconfounding with multiple treatments, we rely on a similar,
> but distinctly different set of assumptions. In particular, existing works on sequential deconfounding (with multiple treatments)
> assume "sequential single strong ignorability" (see Assumption 3 or Eq. 6 in Bica et al., 2020b). This assumption requires that, at every timestep, the unobserved confounder influences at least two treatments. In our case, instead of "sequential single strong ignorability", we assume "time-invariant unobserved confounding". This is needed, since we do not have multiple treatments to leverage, but
> we leverage the sequential treatment assignment. In order to clarify this, we explained the differences between the existing deconfounding works in-depth in the introduction (see revised version). In particular, we explain that existing sequential
> deconfounder works assume "sequential single strong ignorability", whereas our work assumes "time-invariant unobserved confounding".
>
>
> 2. There are two reasons why the existing theory with multiple treatments is not applicable to the setting with a single treatment:
>
> i. Existing work assumes "sequential single strong ignorability", which requires that, at every timestep, the unobserved
> confounder influences at least two treatments. In case there is only a single treatment, this assumption holds never true.
>
> ii. Existing work leverages the dependency between multiple treatments to infer a substitute confounder (via a factor model). However, in the case of a single treatment, there is no dependency between different treatments that could be leveraged.
>
> We clarified these reasons in the introduction of our paper (see revised version).
>
>
> 3. We introduce the random variable Z at the beginning of section "4.1 Step 1: Deconfounding". In particular, the random variable Z is a latent variable inferred by the CMM. Theorem 1 then shows that using this random variable as a substitute for the unobserved confounders U, yields an unbiased estimate for the treatment responses. As such, the random variable Z is valid substitute for the unobserved confounders.
>
> 4. The limitation of the CMM is that it requires the (conditional) treatment assignment to be Markovian. This means that the current treatment assignment depends only on the last treatment assignment, but not on any treatment assignments prior to this. Many established methods for modeling medical decision-making are based on Markovianity, which may be realistic in many applications. However, it may be restrictive in settings in which Markovianity does not hold true. To clarify this, we discuss this limitation in the conclusion of our paper (see revised version) and propose future research to address this.
>
> 5. The static deconfounding works, which are based on multiple treatments, were extended to the time-series setting by Bica et al. 2020b. However, it is not straightforward to extend it to the time-series setting, in which there is only a single treatment (see our comment on 2.). We included this clarification in the introduction (see revised version).
>
> 6. Rubin, 1984 introduces the concept of posterior predictive checks (PPCs). Wang and Blei 2019 propose predictive checks in the context of deconfounding by combining concepts around these PPCs (Rubin 1984), PPCs with realized discrepancies (Gelman, Meng, and Stern 1996), PPCs and held-out data (Gelfand, Dey, and Chang 1992; Ranganath and Blei 2019), and stage-wise checks for hierarchical models (Dey et al. 1998; Bayarri and Castellanos 2007). These predictive checks for deconfounding were then extended to the time-series setting by Bica et al. 2020b. We use the same predictive checks in our work to guide the user to decide whether the model captures the distribution of assigned treatments well.
>
> 7. Note that our theory holds true for any model that infers the latent variable Z in the CMM: it does not restrict the class of models that can be used. As such, a different model class than Gaussian processes can be chosen if the true function cannot be learned with Gaussian processes.

---

> > ### Author Response · Authors · 2022-08-07
> > **Response to Reviewer hN8v**
> >
> > 8. The notion of "unbiased estimation of ITEs" is to be understood as in the causal inference literature. That is, the estimation is unbiased if the conditional expectation $\mathbb{E}[Y\mid \bar{A}_t = \bar{a}_t, \bar{X}_t = \bar{x}_t]$, equals the quantity of interest, i.e., $\mathbb{E}[Y(\bar{a}_t)\mid \bar{X}_t = \bar{x}_t]$. We explain this in section "3 Problem Setup". While the estimation of the conditional expectation, $\mathbb{E}[Y\mid \bar{A}_t = \bar{a}_t, \bar{X}_t = \bar{x}_t]$, can be biased as well (for instance, due to choosing the wrong model class), this is a statistical inference problem rather than a causal inference problem. Hence, we use the notion "unbiased" in a causal inference sense rather than a statistical inference sense, since the main goal of our work is causal inference.

---

### Review · Reviewer_YPZ4 · 2022-08-13

**Summary Of Contributions:**

The paper considers an extension of the deconfounder algorithm into sequential settings, where at each time step there is only one treatment at each time step. The paper proposes to leverage time-invariant confounding to unconver unobserved confounders. If there exists unobserved confounders not captured by the substitute confounder, the treatment assignment cannot be Markov. For example, if some unobserved confounder U is not captured by Z, then A_3 would not only depend on A_2, but also A_1 due to the unobserved U inducing correlations among A_1, A_2, A_3. The paper demonstrate the idea on an instantiation of SeqGPLVM and study the empirical performance on a simulated study and a real world study on MIMIC-III.

**Requested Changes:**

+ Correct Theorem 1 to make formal causal identification argument. Write down the causal quantity as a functional of the observational data distribution only, with all the necessary assumptions required for causal identification.

+ Perform sensitivity analysis for potential violation of both sequential ignorability and pinpointability in all empirical studies.

**Strengths And Weaknesses:**

Strengths:

+ The paper extends the conditional independence argument to the Markov argument with time-invariant confounding.
+ The paper demonstrated the proposal on real world MIMIC-III data and simulated data for validation.

Weaknesses:

+ Theorem 1 is not sufficient for causal identification of Y(a) because Z_t is a latent variable. To make any causal claims, one needs to establish causal identification: how the causal quantity e.g. Y(a) can be written down as a functional of the observational data distribution P(Y, A_{1:t}, X_{1:t}). Theorem 1 by itself is not sufficient for any causal claims due to the unobservability of Z_t. One requires further assumptions to make formal causal claims, e.g. following the theoretical argument in Wang and Blei (2020).

+ Since the estimated Z_t must be a deterministic function of A_{1:t} and X_{1:t} (as Z_t is estimated from A_{1:t}, X_{1:t}), there will be further positivity issues in estimating E[Y|A_{1:t}, X_{1:t}), \hat{Z}_{1:t}] (D’Amour, 2019). One requires additional assumptions or refinement of causal queries to address these positivity issues, e.g.Theorem 6-8 in Wang and Blei (2019).

+ Even with these theoretical arguments fixed to have all the necessary assumptions for formal causal identification results, these assumptions are hard to satisfy in practice. For example, the consistent subsitute confounder assumption in Wang and Blei (2019) (or equivalently, the pinpointability assumption in Wang and Blei (2020)) is necessary for a formal identification result, but it can't be satisfied---even if adapted to the sequential deconfounder---if the number of time steps T is finite. One requires T be infinite to potentially satisfy these assumptions. When these causal identification assumptions are violated, one have to perform sensitivity analysis to assess the impact of these violations. One shall follow (Franks et al, 2019; Zheng et al, 2021a; Zheng et al, 2021b; Zhang et al, 2021) to perform sensitivity analysis in this setting and properly quantify the uncertainty in these estimates.

Wang, Y., & Blei, D. M. (2019). The blessings of multiple causes. Journal of the American Statistical Association, 114(528), 1574-1596.

D’Amour, A. (2019). Comment: Reflections on the deconfounder. Journal of the American Statistical Association, 114(528), 1597-1601.

Wang, Y., & Blei, D. M. (2020). Towards clarifying the theory of the deconfounder. arXiv preprint arXiv:2003.04948.

Franks, A., D’Amour, A., & Feller, A. (2019). Flexible sensitivity analysis for observational studies without observable implications. Journal of the American Statistical Association.

Zheng, J., D'Amour, A., & Franks, A. (2021a). Bayesian Inference and Partial Identification in Multi-Treatment Causal Inference with Unobserved Confounding. arXiv preprint arXiv:2111.07973.

Zheng, J., D'Amour, A., & Franks, A. (2021b). Copula-based sensitivity analysis for multi-treatment causal inference with unobserved confounding. arXiv preprint arXiv:2102.09412.

Zhang, L., Wang, Y., Schuemie, M., Blei, D., & Hripcsak, G. (2021). Adjusting for unobserved confounding using large-scale propensity scores. arXiv preprint arXiv:2110.12235.

---

### Review · Reviewer_PVeW · 2022-08-23

**Summary Of Contributions:**

The paper proposes the Sequential Deconfounder method to estimate individual treatment effects over time when there exist latent confounders, even in one single assigned treatment case. Gaussian process latent variable model is extended to the Sequential Deconfounder and connects the outcome model with the individual treatment effect estimation. Both theoretical and experiments demonstrate the efficacy of the proposed method.

**Broader Impact Concerns:**

Not applicable.

**Requested Changes:**

1. I am concerned with the identifiability of the substitute of the latent confounders. It seems difficult to let me convince all the information of latent confounders can be captured by Z by the proposed method.

2. The part about how to infer latent variables Z is unclear to me. It would be better to provide more details on how to infer the posterior distribution of the substitute Z.

3. I am concerned that the performance is influenced by the chosen kernel for SeqGPLVM. If the chosen kernel does not match the joint space of real data, the method fails to estimate the correct individual treatment effects.

**Strengths And Weaknesses:**

Strengths:
1. The setting of this paper is interesting and close to the real scenario.
2. The proposed method can be applied in the case of a single assigned treatment over time, with the time-invariance latent confounder assumption.

Weakness:
1. Some parts are unclear and need to provide more details.
2. The SeqGPLVM may be sensitive to the kernel chosen.

---

### Decision · Action_Editors · 2022-09-25

**Recommendation:** Reject

**Comment:**

The paper extends deconfounding to sequential settings and considers the more real and challenging scenario where there is only one treatment at each time step. While the proposed problem setting is interesting, it lacks theoretical guarantee on the identifiability of causal effect under unmeasured confounding. It is unclear whether the causal effect is identifiable with the latent variable z as a surrogate for unmeasured confounders. As a result, the theoretical correctness of the proposed method is not guaranteed. Thus, I recommend rejection of this paper.